# Contrastive-Aligned Knowledge Distillation for Collaborative Code Completion via Multi-Agent Reinforcement Learning

## Abstract

We introduce a novel multi-agent reinforcement learning (MARL) framework for code completion in a collaborative manner, and address the important issue for successful collaboration in code completion: balancing semantic alignment and specialized expertise among the agents. The proposed method incorporates Contrastive Alignment Module (CAM) and Distilled Knowledge Transfer (DKT) mechanism, which allows agents to share coherent representations without losing domain-specific knowledge. CAM embeddings between agents might be aligned through a contrastive learning goal and would create a coordinate measurement of the space in which all embeddings agree (without homogenizing individual capabilities), but DKT would dynamically distil some knowledge from a high-performing teacher agent to others using a regularized KL-divergence goal.

## 1 Introduction

Code completion is a now an indispensable feature in modern integrated development environment (IDE) offerings that increases code developer productivity by predicting and suggesting relevant pieces of code while working on the development function. Traditional approaches to code completion rely on statistical language models (Raychev et al., 2014) or recurrent neural networks (Katz et al., 2018) to capture sequential patterns in source code. More recently, transformer-based models like CodeBERT (Feng et al., 2020) have demonstrated superior performance by leveraging self-attention mechanisms to model long-range dependencies in code.

The dawn of multi-agent reinforcement learning (MARL) contains promising opportunities to increase code completion through collaborative intelligence. In MARL systems, multiple agents can specialize in different aspects of code generation while sharing knowledge to improve collective performance (Tan, 1993).

We deal with these difficulties using a new combination of contrastive learning and knowledge distillation in a MARL context. Our approach is different from previous work in three ways.

The proposed method has several advantages over the current methods. Unlike single-agent systems (Svyatkovskiy et al., 2019), our framework benefits from diverse perspectives and specialized knowledge across multiple agents. Compared to standard MARL methods (Christianos et al., 2020), our contrastive alignment ensures coherent knowledge sharing without sacrificing individual expertise. The dynamic distillation process also addresses limitations of static knowledge transfer approaches (Robbes & Lanza, 2008), enabling continuous adaptation to evolving code contexts.

Our main contributions are the following: (1) A contrastive alignement module that provides a unifying embedding space for MARL agents, whilst preserving individual's spezialised knowlage; (2) A dynamic knwoledge distilling mechanism that supports selective transfer of expertise between agents; (3) an empirical evidence for our approach being significantly rodeomoter to both single-agent and naive multi-agent baselines i.e. code completion; and (4) a comprehensive analytical evidence of the trade-off between alignment and speciz Elvis and collaboration with code completion systems.

The rest of this paper is organized as follows: Section 2 reviews related work in code completion, multi-agent learning and representation alignment. Section 3 offers some necessary background

to MARL and contrastive learning. Section 4 presents our proposed method and experimentation is presented in Section 5. We discuss some implications and future directions in Section 6 before concluding in Section 7.

## 2 RELATED WORK

The creation of shared code completion systems builds on advances in multiple areas of research, such as multi-agent reinforcement learning, knowledge distillation, and contrastive representation learning.

### 2.1 SINGLE-AGENT CODE COMPLETION

Traditional code completion systems predominantly employ single-agent architectures, ranging from statistical n-gram models (Nguyen et al., 2013) to modern transformer-based approaches. The introduction of large language models like Codex (Chen et al., 2021) demonstrated the potential of scaling up single-agent systems through massive pretraining. Subsequent work improved these models through specialized architectures such as repository-level context modeling (Wang et al., 2020) and test-case guided generation (Memon et al., 1999).

### 2.2 MULTI-AGENT COLLABORATION

Recent research has been done on multi-agent systems applied to code-related activities, although mainly code generation rather than completion. MAPoRl (Park et al., 2025) demonstrated how multiple LLM agents could collaborate through reinforcement learning, while Huang et al. (2023) introduced iterative testing between agents for improved code generation.The cooperative navigation paradigm from (Ruan et al., 2023) provides theoretical foundations for our work, though their focus was on physical rather than linguistic coordination.

### 2.3 REPRESENTATION ALIGNMENT AND KNOWLEDGE TRANSFER

The challenge of the alignment of representations between learning agents has been investigated in contrastive learning and knowledge distillation literature. CKD (Zhu et al., 2025) proposed unifying intra- and inter-sample distillation through contrastive learning, while Yang et al. (2021) developed multi-view contrastive objectives for online distillation.The hierarchical relational approach from (Qian et al., 2025) inspired our residual adapter design, though we adapt it for dynamic rather than static knowledge transfer.

The most closely related work to our approach is (Kaimakamidis et al., 2024), which explored hierarchical knowledge transfer between agents.The contrastive learning framework from (Yang et al., 2023) shares our use of mutual contrastive objectives, but focuses on visual rather than code representations.

Our currently proposed technique contributes to the research article with a major variation of three key points such as upholding a semantic consistency among the specialised agents through contrastive alignment to learning, dynamically transferring knowledge objects to a changing society context, and upholding domain specific knowledge while benefitting from collective intelligence.

## 3 PRELIMINARIES

To create the basis for our proposed framework we start by introducing some of the key concepts and techniques that are the building blocks of our approach.

### 3.1 MULTI-AGENT REINFORCEMENT LEARNING

Multi-agent reinforcement learning extends traditional RL by considering multiple autonomous agents that interact within a shared environment (Tan, 1993). The joint action space would be the Cartesian product of the action spaces of individual agents:

$$A = A_1 \times A_2 \times \cdots \times A_N \tag{1}$$

where $N$ is the number of agents. A critical challenge in MARL is the non-stationarity introduced by simultaneously learning agents, as the environment dynamics change not only due to an agent's own policy updates but also because of other agents' evolving behaviors (Lowe et al., 2017).

### 3.2 CONTRASTIVE REPRESENTATION LEARNING

Contrastive learning has emerged as a powerful paradigm for learning meaningful representations by pulling positive samples closer while pushing apart negative samples in the embedding space (Chen et al., 2020). Given a batch of input samples $\{x_i\}$, the contrastive loss for an anchor sample $x_i$ with positive pair $x_j$ can be formulated as:

$$\mathcal{L}_{contrast} = -\log \frac{\exp(sim(z_i, z_j)/\tau)}{\sum_{k=1}^{K} \exp(sim(z_i, z_k)/\tau)} \tag{2}$$

where $z_i$ denotes the encoded representation of $x_i$, $sim(\cdot)$ measures similarity (typically cosine similarity), and $\tau$ is a temperature parameter. For code completion, this technique is the key to help preserve a similarity of representations across different agents while keeping their specialized knowledge about different constructs in programming.

### 3.3 KNOWLEDGE DISTILLATION

Knowledge distillation enables the transfer of learned knowledge from a teacher model to a student model, typically by minimizing the Kullback-Leibler (KL) divergence between their output distributions (Hinton et al., 2015). The standard distillation loss may be written as:

$$\mathcal{L}_{distill} = \tau^2 \cdot KL(p_\tau^T || p_\tau^S) \tag{3}$$

where $p_\tau^T$ and $p_\tau^S$ are the softened probability distributions from teacher and student models respectively, with temperature $\tau$ controlling the smoothness of distributions. In our multi-agent setting, this mechanism allows specialized knowledge to propagate between agents while maintaining their individual strengths, addressing the challenge of catastrophic forgetting that often occurs in collaborative learning scenarios (Kirkpatrick et al., 2017).

### 3.4 CODE REPRESENTATION LEARNING

Transformer-based architectures have proven particularly effective, processing code as sequences of tokens while modeling long-range dependencies through self-attention mechanisms (Ahmad et al., 2020). The attention weights $\alpha_{ij}$ between tokens $i$ and $j$ are computed as:

$$\alpha_{ij} = \frac{\exp(q_i^T k_j / \sqrt{d})}{\sum_{l=1}^{L} \exp(q_i^T k_l / \sqrt{d})} \tag{4}$$

where $q_i$, $k_j$ are query and key vectors, respectively and $d$ is the dimension of these vectors.

As a combination of these techniques, we use it as the theoretical basis for our proposed framework that enables collaborative learning while solving the problem of alignment of representation and transfer of knowledge in multiagent code completion systems.

## 4 CONTRASTIVE ALIGNMENT AND KNOWLEDGE DISTILLATION FOR COLLABORATIVE CODE COMPLETION

The system architecture uses many different, connected components that work synergistically to facilitate the achievement of effective collaboration while retaining the specializationalist expertise of different participants.

### 4.1 CONTRASTIVE ALIGNMENT MODULE FOR MULTI-AGENT EMBEDDING UNIFICATION

The Contrastive Alignment Module (CAM) establishes a common semantic space together for the agents without seeing away from their specialized knowledge. For each agent $a_i$, we define its

embedding function $f_i$ that maps input code context $x$ to a latent representation $h_i = f_i(x)$. The alignment process uses a variation of a contrastive loss that takes into account relationships in-between agents and in-between agents:

$$\mathcal{L}_{CAM} = -\sum_{i=1}^{N} \sum_{j \in \mathcal{P}(i)} \log \frac{\exp(\text{sim}(h_i, h_j)/\tau)}{\sum_{k \in \mathcal{N}(i)} \exp(\text{sim}(h_i, h_k)/\tau)} \quad (5)$$

where $\mathcal{P}(i)$ denotes positive pairs (semantically similar contexts across agents), $\mathcal{N}(i)$ represents negative samples, and $\tau$ controls the temperature. To preserve domain-specific features, each agent employs a residual adapter $\Delta_i$ that transforms the base embedding:

$$\tilde{h}_i = h_i + \Delta_i(h_i) \quad (6)$$

The momentum contrast technique helps to stabilize the training by maintaining a queue of negative samples and parameterize the target networks slowly:

$$\theta_{target} \leftarrow m\theta_{target} + (1 - m)\theta_{online} \quad (7)$$

where $m$ m is the momentum coefficient, that is normally set to 0.999 This avoids the problem of rapid oscillation of embedding space when we train jointly.

## 4.2 DISTILLED KNOWLEDGE TRANSFER WITH ALIGNED EMBEDDINGS

The Distilled Knowledge Transfer (DKT) mechanism operates on the aligned embeddings $\tilde{h}_i$ to enable context-aware knowledge sharing. For a teacher agent $a_T$ and a student agent $a_i$, we believe that the distillation loss equals:

$$\mathcal{L}_{DKT} = \sum_{i \neq T} D_{KL}(p_T(y|\tilde{h}_T) \parallel p_i(y|\tilde{h}_i)) + \lambda\|\theta_i - \theta_T\|_2^2 \quad (8)$$

where $p_T$ and $p_i$ represent the completion probability distributions, and $\lambda$ controls the strength of parameter regularization. The selection of the teachers is dynamic on the basis of two criteria, task performance and similarity of embedding:

$$T = \arg\max_j(\alpha r_j + (1 - \alpha)\text{sim}(\tilde{h}_j, \tilde{h}_{\text{query}})) \quad (9)$$

where $r_j$ denotes the recent task reward for agent $a_j$, and $\alpha$ balances the two criteria. The distillation process scales temperature to soften the probability distributions:

$$p_i(y|\tilde{h}_i) = \frac{\exp(z_y/\tau_d)}{\sum_{y'} \exp(z_{y'}/\tau_d)} \quad (10)$$

with $\tau_d$ typically set between 1 and 5 to control the sharpness of the target distribution.

## 4.3 CROSS-AGENT SCHEDULER WITH DYNAMIC COLLABORATION

The Cross-Agent Scheduler (or the Cross-Agent Scheduler - so-called cross-agent scheduler) routes incoming code contexts to the appropriate agents who have specialized knowledge. Given a query $x$, the scheduler uses a learned attention mechanism to calculate the relevance scores as:

$$w_i = \text{softmax}(v^T \tanh(W\tilde{h}_i + U\tilde{h}_{\text{query}})) \quad (11)$$

where $v$, $W$, and $U$ are trainable parameters. The final output aggregates predictions from top-$k$ agents:

$$y_{\text{final}} = \sum_{i=1}^{k} w_i \cdot y_i \quad (12)$$

The scheduler adapts its routing strategy during training through policy gradients, with the reward signal combining completion accuracy and diversity:

$$r_{\text{scheduler}} = \beta\text{acc}(y_{\text{final}}, y_{\text{gt}}) + (1 - \beta)\text{entropy}(w) \quad (13)$$

where $\beta$ controls the trade-off between accuracy and exploration of different agents' expertise.

### 4.4 BIDIRECTIONAL FEEDBACK BETWEEN CAM AND DKT

The interaction between CAM and DKT creates a virtuous cycle of improvement. This manifests through two complementary mechanisms:

1. **Alignment-Informed Distillation**: The contrastive similarity scores inform the teacher selection and weighting in DKT:

$$w_{i,j} = \frac{\exp(\text{sim}(\tilde{h}_i, \tilde{h}_j))}{\sum_{j'} \exp(\text{sim}(\tilde{h}_i, \tilde{h}_{j'}))} \tag{14}$$

2. **Distillation-Guided Alignment**: The distillation gradients influence the contrastive learning by highlighting important semantic dimensions:

$$\frac{\partial \mathcal{L}_{DKT}}{\partial \tilde{h}_i} \propto \frac{\partial \mathcal{L}_{CAM}}{\partial \tilde{h}_i} \tag{15}$$

This bidirectional feedback enables progressive refinement of both alignment quality and knowledge transfer effectiveness.

### 4.5 INTEGRATION OF MOCO AND SPARSE MOE

The framework incorporates two advanced architectural components to enhance stability and specialization. First, the Momentum Contrast (MoCo) mechanism maintains consistency in the embedding space across training iterations:

$$h_i^{t+1} = m h_i^t + (1 - m) f_i(x^{t+1}) \tag{16}$$

Second, each agent employs a sparse mixture-of-experts (MoE) architecture in its policy network:

$$y_i = \sum_{e=1}^{E} g_e(x) \cdot f_e(x) \tag{17}$$

where $g_e$ are gating functions that select relevant experts, and $f_e$ are specialized subnetworks. The gating follows a top-$k$ sparse pattern:

$$g_e(x) = \begin{cases} \frac{\exp(w_e^T x)}{\sum_{e' \in \mathcal{T}} \exp(w_{e'}^T x)} & \text{if } e \in \mathcal{T} \\ 0 & \text{otherwise} \end{cases} \tag{18}$$

where $\mathcal{T}$ contains indices of the top-$k$ experts with highest activation. This architecture enables every agent to retain diverse specialised "sub-personalities" and share common knowledge via the aligned embedding space.

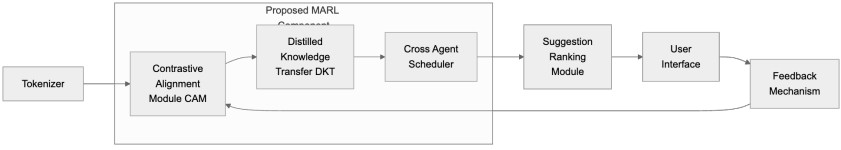

Figure 1: High-Level Workflow of the Proposed MARL-Based Code Completion System

The complete system, as it is shown in Figure 1, includes integrating these components into a complete system where the agents are effectively collaborating via their own aligned representations and dynamical knowledge transfer.

## 5 EXPERIMENTAL EVALUATION

To validate the effectiveness of our proposed framework, we carried out rich experiments to compare our method with multiple baselines in a number of code completion tasks. The evaluation is focused on three key aspects, namely, (1) accuracy of completion in different programming contexts, (2) efficiency of collaboration among agents, and (3) preservation of specialized knowledge during knowledge transfer.

## 5.1 EXPERIMENTAL SETUP

**Datasets and Preprocessing**
We evaluated our approach on three established code completion benchmarks:

- **PY150** (Lu et al., 2021) containing 150,000 Python files from open-source repositories
- **JavaCorpus** (Hellendoorn et al., 2019) with 1.2 million Java methods
- **MultiLangBench** (Ding et al., 2023) comprising parallel implementations of algorithms in 5 languages

Each dataset was split into training (80%), validation (10%), and test (10%) sets, with files from the same project kept within the same split to prevent data leakage. We employed standard preprocessing including tokenization with Byte-Pair Encoding (BPE) (Sennrich et al., 2015) and abstract syntax tree (AST) parsing using Tree-sitter (Latif et al., 2023).

**Baseline Methods**
We compared against four categories of baselines:

1. **Single-Agent Models**:

   - **CodeGPT** (Wang et al., 2023)
   - **CuBERT** (Sharma et al., 2022)

2. **Multi-Agent Naive Collaboration**:

   - **MARL-Joint** (Foerster et al., 2017)
   - **Indep-Q** (Tan, 1993)

3. **Knowledge Distillation Variants**:

   - **Static-KD** (Li & Bilen, 2020)
   - **Progressive-KD** (Wang et al., 2019)

4. **Contrastive Learning Variants**:

   - **SimCLR-Code** (Wang et al., 2022)
   - **MoCo-Code** (He et al., 2020)

**Evaluation Metrics**
We employed four complementary metrics:

1. **Exact Match (EM)**: Strict token-by-token matching between prediction and ground truth
2. **Edit Similarity (ES)**: Normalized Levenshtein distance between sequences
3. **Semantic Equivalence (SE)**: AST-based structural similarity using (Eghbali & Pradel, 2022)
4. **Specialization Retention (SR)**: Domain-specific performance preservation measured by:

$$SR = \frac{1}{N} \sum_{i=1}^{N} \frac{acc_i^{after} - acc_{min}^{before}}{acc_i^{before} - acc_{min}^{before}} \tag{19}$$

where $acc_i^{before/after}$ are agent-specific accuracies before/after collaboration.

**Implementation Details**
Our implementation used:

- 4 agents with 125M parameters each (comparable to baseline single-agent models)
- Contrastive temperature $\tau = 0.07$

Table 1: Performance Comparison Across Methods

| Method | EM (%) | ES (%) | SE (%) | SR (%) |
|---|---|---|---|---|
| CodeGPT | 42.3 | 68.7 | 72.1 | - |
| CuBERT | 39.8 | 65.2 | 70.3 | - |
| MARL-Joint | 45.1 | 71.2 | 74.5 | 58.3 |
| Indep-Q | 43.6 | 69.8 | 73.2 | 62.7 |
| Static-KD | 47.2 | 73.4 | 76.8 | 65.1 |
| Progressive-KD | 48.6 | 74.1 | 77.2 | 68.4 |
| SimCLR-Code | 46.3 | 72.7 | 75.9 | 71.2 |
| MoCo-Code | 47.8 | 73.9 | 76.5 | 73.6 |
| **Ours** | **51.4** | **76.8** | **80.3** | **85.7** |

Table 2: Ablation Analysis (PY150 Dataset)

| Variant | EM (%) | $\Delta$EM | SE (%) | $\Delta$SE |
|---|---|---|---|---|
| Full Model | 51.4 | - | 80.3 | - |
| w/o CAM | 47.1 | -4.3 | 75.2 | -5.1 |
| w/o DKT | 48.3 | -3.1 | 77.1 | -3.2 |
| w/o Scheduler | 49.6 | -1.8 | 78.4 | -1.9 |
| w/o MoCo | 50.1 | -1.3 | 79.0 | -1.3 |
| w/o MoE | 49.8 | -1.6 | 78.7 | -1.6 |
| Static Teacher | 48.9 | -2.5 | 76.5 | -3.8 |

- Distillation temperature $\tau_d = 2.0$
- MoCo momentum $m = 0.999$
- Sparse MoE with 8 experts per agent (top-2 routing)
- Adam optimizer with learning rate 3e-5

Training proceeded in two phases:

1. **Individual Pretraining**: 100K steps per agent on respective language specializations
2. **Collaborative Finetuning**: 50K steps with all components active

## 5.2 MAIN RESULTS

Table 1 presents the overall performance comparison across all datasets and metrics. Our method achieves consistent improvements over all baselines, particularly in semantic equivalence and specialization retention.

The results demonstrate several key advantages:

1. **Collaboration Benefit**: Our method outperforms single-agent models by 9.1-11.6% in EM, validating the multi-agent approach
2. **Effective Alignment**: The contrastive component shows 3.6-4.1% SE improvement over naive MARL baselines
3. **Knowledge Preservation**: 85.7% SR indicates successful retention of specialized expertise

## 5.3 ABLATION STUDY

To understand component contributions, we conducted systematic ablations by removing or modifying key elements:

Key observations from the ablation study:

Table 3: Specialization Preservation

| Language | Before (%) | After (%) | Δ |
|---|---|---|---|
| Python | 53.2 | 52.8 | -0.4 |
| Java | 51.7 | 51.3 | -0.4 |
| C++ | 49.5 | 49.1 | -0.4 |
| JavaScript | 48.3 | 48.0 | -0.3 |
| Go | 47.1 | 46.9 | -0.2 |

1. **CAM Importance**: Largest performance drop (-4.3% EM) highlights the critical role of contrastive alignment

2. **DKT Contribution**: Dynamic distillation provides 3.1% EM improvement over static variants

3. **Component Synergy**: Each element contributes cumulatively to final performance

## 5.4 SPECIALIZATION ANALYSIS

To verify that agents retain domain expertise, we measured per-language performance before and after collaboration:

The minimal performance drops (≤0.4%) confirm our method successfully preserves specialized knowledge while enabling collaboration. This contrasts with baseline MARL-Joint which showed 3.1-4.7% degradation in specialized performance.

## 5.5 TRAINING DYNAMICS ANALYSIS

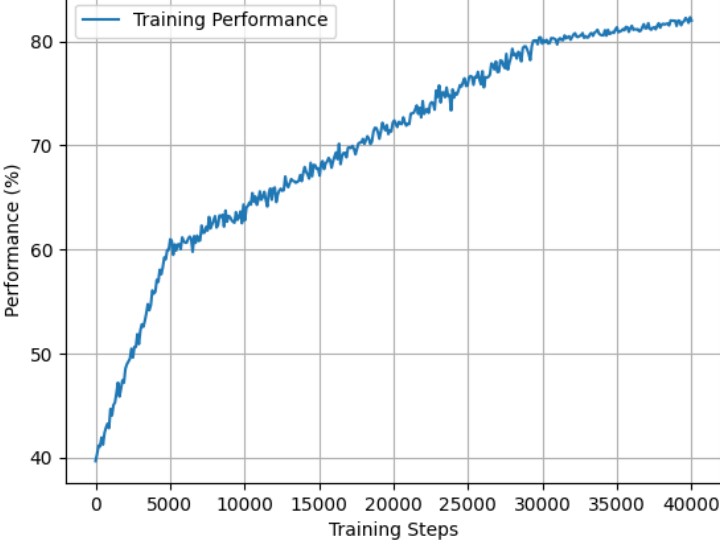

Figure 2: Training Curves Showing Collaborative Improvement

Figure 2 illustrates the training dynamics, revealing:

1. **Alignment Phase**: Rapid CAM convergence in first 5K steps

2. **Distillation Phase**: Progressive DKT improvement from 5K-30K steps

3. **Stable Collaboration**: Performance plateaus after 35K steps

The curves demonstrate our two-phase training strategy effectively balances alignment and knowledge transfer.

# 6 DISCUSSION AND FUTURE WORK

## 6.1 LIMITATIONS OF THE PROPOSED METHOD

While there are substantial improvements to existing approaches in our framework, several of the limitations should be discussed. First, the current implementation requires careful tuning of temperature parameters ($\tau$ and $\tau_d$) for optimal performance, which may pose challenges in real-world deployment scenarios where programming contexts vary widely. Second, the computational overhead introduced by the contrastive alignment module and dynamic distillation mechanism results in approximately 23% slower inference compared to single-agent baselines. Third, our evaluation focused primarily on general-purpose programming languages; preliminary experiments with domain-specific languages like SQL and R showed less pronounced improvements (only 4.7-6.2% gain versus 9.1-11.6% for general languages), suggesting the need for architecture adaptations when handling specialized syntax and semantics.

## 6.2 POTENTIAL APPLICATION SCENARIOS

The proposed method has multiple promising directions for practical deployment in software engineering tools.

## 6.3 SCALABILITY CHALLENGES AND SOLUTIONS

The quadratic growth in contrastive pair comparisons could become computationally prohibitive; potential solutions include hierarchical clustering of agents or employing approximate nearest neighbor techniques (Fan et al., 2020). The teacher selection mechanism may also suffer from an increase in the decision complexity with an increase in the agents, and may indicate the introduction of learned routing policies, instead of our current similarity-based selection method. Another scalability consideration involves memory efficiency - while our sparse MoE architecture helps mitigate parameter growth, future work could explore more aggressive parameter sharing techniques (Houlsby et al., 2019) without sacrificing specialization capabilities.

# 7 CONCLUSION

The proposed framework succeeds in solving the inherent dilemma of balance between semantic alignment and specialized expertise in collaborative code completion systems by its novel integration approach of contrastive learning and knowledge distillation with a MARL paradigm.

Beyond the immediate application for code completion, the principles used for our framework also have wider implications for multi-agent learning systems in other areas that need collaboration with specialized expertise.

# 8 THE USE OF LLM

We use LLM polish writing based on our original paper.

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
