# OpenReview forum: "Contrastive-Aligned Knowledge Distillation for Collaborative Code Completion via Multi-Agent Reinforcement Learning"
_ICLR.cc/2026/Conference — Submitted to ICLR 2026_

### Official Review · Reviewer_Vorc · 2025-10-18

**Soundness:** 1
**Presentation:** 1
**Contribution:** 1
**Rating:** 0
**Confidence:** 2

**Summary:**

The paper claims to use Multia-agent Reinforcement Learning to improve code completion. The method combines Contrastive Alignment module and Distilled Knowledge Transfer so that embeddings between agents are aligned

**Strengths:**

The paper is really bad, it's hard to find a strength.

**Weaknesses:**

The paper is very badly written, from careless typos that show the paper was barely revised (e.g. "spezialised knowlage"; "speciz Elvis"), to a lack of cohesive structure between the different sections and terrible use of equations (virtually the same equation is rewritten in slightly different way in (2), (4), (5), (10), (14), (18)). When I finished reading the paper I sincerely couldn't have explained to someone else what is the main proposal of the paper and how it was implemented in the experimental evaluation.

I am not even convinced the authors know what Multiagent Reinforcement Learning is (which is what is claimed to be the main contribution of the paper). The background on that simply says that in a MARL problem each agent has its own actions - with absolutely no description of the other elements of a Multiagent MDP. Most importantly, there isn't a single sentence in the whole paper dedicated to explaining what is the state-action-reward space for the code completion domain, which directly means that the reader can't have a minimal idea of how the baselines were implemented.

The paper has to be entirely rewritten - and I am not even convinced it makes any sense at all to use MARL in this domain. Whatever the authors did in their experiment, it is not MARL.

**Questions:**

No Question, paper is a clear reject.

---

### Official Review · Reviewer_hxQN · 2025-10-27

**Soundness:** 2
**Presentation:** 2
**Contribution:** 2
**Rating:** 2
**Confidence:** 3

**Summary:**

The paper proposes a multi-agent reinforcement learning (MARL) framework for code completion that combines:

	1.	Contrastive Alignment Module (CAM) to align latent representations across agents

	2.	Distilled Knowledge Transfer (DKT) for dynamic “teacher → student” imitation

	3.	A routing scheduler using soft attention over agents

	4.	MoCo queue + Sparse MoE experts inside agents for specialization

Claims: improved completion metrics (EM/ES/SE) and stronger “specialization retention” compared to single-agent and naive multi-agent baselines (Tables 1–3).

**Strengths:**

1. Acknowledges the alignment vs. specialization dilemma in collaborative code modeling.

2. The idea of using contrastive similarity to guide teacher selection is reasonable in theory (Eq.9).

3. Ablation tables show some component contributions, albeit marginal.1.

**Weaknesses:**

1. All components are known and simply combined: multi-agent code modeling (MAPoRL: Park et al., 2025; AgentCoder: Huang et al., 2023), contrastive alignment (MoCo-Code, SimCLR-Code), collaborative KD (CKD: Zhu et al., 2025; Yang et al., 2023), MoE for specialization (CodeT5+ family). The paper does not introduce a new algorithmic principle.

2. Eq.(5) is SimCLR-style contrastive loss with changed notation. Eq.(8) is standard KD with L2 regularization. Eq.(15) claims gradient synergy without theoretical justification or analysis of conflicting gradients.

3. CodeGPT and MARL-Joint are not competitive code completion baselines in 2025/2026. Strong baselines such as CodeT5+, StarCoder, StarCoder2, and DeepSeek-Coder are missing. Claims of state-of-the-art performance are unsubstantiated.

4. The Specialization Retention metric (Eq.19) is ad-hoc and does not assess disentanglement, cross-agent interference, or negative transfer.

5. The environment structure, reward horizon, and credit assignment in MARL are not clearly defined. The scheduler could likely be replaced by supervised routing with no RL.

6. Code completion requires compilation, runtime correctness, and human evaluation. Token-level exact match is insufficient for semantic code quality.

7. The authors acknowledge 23\% slower inference and quadratic scaling in contrastive components. Without practical feasibility, the method is unlikely to see adoption.

**Questions:**

1.	How is this a MARL setting if each agent predicts code independently and no joint action sequences are evaluated?

	2.	Why hide comparisons with CodeT5+/StarCoder that would outperform your 125M agents with zero collaboration?

	3.	Where are human evals, compile success, static analysis correctness? Token-level edit distance ≠ sanity.

	4.	How do you guarantee expertise preservation, rather than embedding homogenization?

	5.	How is teacher selection (Eq.9) not reinforcing a single dominant policy, leading to collapse?

	6.	Why is Eq.(15) even true? Show gradient alignment or stop making causal claims.

	7.	Where is the analysis of conflicts when CAM and DKT gradients oppose each other?

	8.	How does performance scale >4 agents? Complexity grows quadratically in contrastive pairs — is that workable?

---

### Official Review · Reviewer_3dJF · 2025-10-29

**Soundness:** 2
**Presentation:** 1
**Contribution:** 2
**Rating:** 2
**Confidence:** 4

**Summary:**

This paper proposes a MARL framework for collaborative code completion, introducing two core components:
1. A Contrastive Alignment Module (CAM) to align semantic embeddings across agents while maintaining their specialized knowledge.
2. A Distilled Knowledge Transfer (DKT) mechanism for dynamic teacher-student knowledge sharing based on task performance and embedding similarity.

The model further integrates a Cross-Agent Scheduler, Momentum Contrast (MoCo), and a Sparse Mixture-of-Experts (MoE) architecture to enhance stability and specialization. Experiments on PY150, JavaCorpus, and MultiLangBench*datasets show significant improvements in code completion accuracy and specialization retention over baselines such as CodeGPT, MARL-Joint, and MoCo-Code.

**Strengths:**

- The topic (collaborative code completion) is timely and relevant to MARL and software engineering.
- The authors conduct experiments across several datasets and include ablations.

**Weaknesses:**

1. My main concern is that this paper lack of genuine novelty. The method combines existing paradigms (contrastive learning + KD + MARL) without substantive conceptual innovation for me.
2. No formal analysis supports why contrastive alignment and distillation jointly improve collaboration.
4. The writing is poor.

**Questions:**

1. How exactly does the CAM–DKT interaction differ from prior contrastive distillation methods (e.g., CKD)?
2. Have the authors analyzed training stability and convergence beyond qualitative curves?
3. Can the system scale beyond four agents without exponential computational cost?
4. Why is there no comparison with recent large-scale collaborative LLM frameworks (e.g., MAPoRL 2025)?
5. What ablation evidence directly isolates the contribution of bidirectional CAM–DKT coupling?

---

### Official Review · Reviewer_v45Z · 2025-11-11

**Soundness:** 1
**Presentation:** 1
**Contribution:** 1
**Rating:** 2
**Confidence:** 4

**Summary:**

This paper proposes a multi-agent RL (MARL) algorithm for code completion. They develop a novel algorithm where there are 2 key novel components: a contrastive alignment module which is uses to align the internal representations of different agents and a distilled knowledge transfer module that helps the agents share knowledge, which I believe has the same motivation as papers like Distral [1], which aim to distill knowledge between different components of an algorithm. They show that this enables better code completion that baseline methods, ablate the various components, and show a figure that they argue shows that collaboration is improving.

[1] Distral: Robust Multitask Reinforcement Learning

**Strengths:**

The results are good and the idea is sensible. Distral was a very powerful Distillation and transfer learning algorithm and its not hard to imagine that ideas like this would improve MARL algorithms for code completion.

It's also good that you compare against many baselines.

**Weaknesses:**

there are many types. e.g.
- "A dynamic knwoledge"
- "significantly rodeomoter to both singleagent and naive multi-agent baseline". "rodeomoter"?

the related work needs work for explaining how this contribution fits within previous context. They describe prior work by they don't explain *what* is different about their algorithm of method

they don't really motivate using a MARL algorithm instead of a single agent. They only say that this will enable agents with specialized knowledge but its really not obvious that this will be better. I think an *experiment* that shows this is critical for this argument.

many things were unclear:
- what is "semantically similar contexts across agent"?
- why does a residual adaptive $\Delta_i$ preserve domain-specific features? why do you want this?
- I don't have a sense of what the algorithm does *overall* and why. I don't quite understand why you're distilling knowledge transfer?
- you introduce "CROSS-AGENT SCHEDULER WITH DYNAMIC COLLABORATION" in section 4.3 for the first time and I have no idea why you're introducing this and how its related to your algorithm. likewise for 4.4 or 4.5

For the ablations, and for all numbers, what are the errors across multiple seeds? The values are so close that it's hard to compare.

I don't find Figure 2 compelling at all that this shows collaborative improvement. Many MARL papers, e.g. [1], have shown that higher task performance can with less collaboration.

Overall, this paper was challenging to follow. I don't know why you want a MARL system and what you want to distill some knowledge but share other. And I don't have a clear sense of the algorithm. I think you really need an algorithm or a figure that places all the pieces together so that one can understand how they relate to each other

[1] Cross-environment Cooperation Enables Zero-shot Multi-agent Coordination

**Questions:**

why did you compare against CodeGPT instead of GPT5?

---

### Meta-Review · Area_Chair_YRXR · 2025-12-19

**Summary:**

This paper presents a multi-agent reinforcement learning (MARL) approach for code completion, which incorporates two key components: a contrastive alignment module and a distilled knowledge transfer module. The authors demonstrate promising experimental results with their method.

The strengths of this work are summarized below:
1. The topic is timely.
2. The empirical performance reported is promising.

The main weaknesses are as follows:
1. The writing requires significant improvement.
2. The motivation for framing code completion as a multi-agent RL problem is unclear.
3. The claimed novelty appears limited.

**Reviewer Concerns:**

The authors did not response to the reviews.

**Reviewer Scores:**

The authors did not response to the reviews.

---

### Decision · Program_Chairs · 2026-01-26

Reject